# Serum MicroRNAs as Predictors for HCV Progression and Response to Treatment in Pakistani Patients

**DOI:** 10.3390/genes14020441

**Published:** 2023-02-09

**Authors:** Sadia Manzoor, Imran Riaz Malik, Shah Jahan, Muhammad Bilal Sarwar, Asma Bashir, Sulaiman Shams, Abrar Hussain

**Affiliations:** 1Department of Biotechnology, University of Sargodha, Sargodha 40100, Pakistan; 2Department of Immunology, University of Health Sciences, Lahore 54700, Pakistan; 3Food and Biotechnology Research Center, PCSIR Laboratories Complex Lahore, Lahore 54700, Pakistan; 4Department of Biosciences, Faculty of Life Sciences, Shaheed Zulfikar Ali Bhutto Institute of Science and Technology (SZABIST), Karachi 75600, Pakistan; 5Department of Biochemistry, Abdul Wali Khan University Mardan, Mardan 23200, Pakistan; 6Faculty of Life Sciences and Informatics, BUITEMS, Quetta 87300, Pakistan

**Keywords:** hepatitis C, chronic, cirrhosis, HCC, treatment, biomarker

## Abstract

Hepatitis is one of the common liver diseases, imposing a heavy health burden worldwide. Acute hepatitis may develop into chronic hepatitis, progressing to cirrhosis and hepatocellular carcinoma. In the present study, the expression of miRNAs was quantified by real-time PCR, such as miRNA-182, 122, 21, 150, 199, and 222. Along with the control group, HCV was divided into chronic, cirrhosis, and HCC groups. The treated group was also included after the successful treatment of HCV. Biochemical parameters, such as ALT, AST, ALP, bilirubin, viral load, and AFP (HCC), were also evaluated in all of the study groups. We compared the control and diseased groups; these parameters showed significant results (*p* = 0.000). The viral load was high in HCV but was not detected after treatment. miRNA-182 and miRNA-21 were overexpressed with disease progression, while the expression of miRNA-122 and miRNA-199 was increased compared with the control, but decreased in the cirrhosis stage compared with chronic and HCC. The expression of miRNA-150 was increased in all of the diseased groups compared with the control, but decreased compared with the chronic group. We compared the chronic and treated groups and then all of these miRNAs were down-regulated after treatment. These microRNAs could be used as potential biomarkers for diagnosing different stages of HCV.

## 1. Introduction

Hepatitis is one of the most common liver diseases, known as liver inflammation, which imposes a heavy health burden worldwide. Acute hepatitis may develop into chronic hepatitis, or be self-resolving, progressing to hepatocellular carcinoma (HCC) or cirrhosis [1]. The leading causes of hepatitis include metabolism, infection, and autoimmune-related problems [2].

The important pathogen in humans that is responsible for causing hepatocellular carcinoma, liver cirrhosis, and hepatitis is Hepatitis C virus (HCV). It imposes a severe problem related to public health worldwide because it could increase the number of patients chronically infected with HCV that are now at risk of progressive liver disease [3].

As per the record of the World Health Organization, about 71 million people are infected with the Hepatitis C virus worldwide. It is reported that around 399,000 people identified as having Hepatitis C virus convert to liver failure, cirrhosis, and hepatocellular carcinoma (HCC) yearly. For this reason, screening these HCV-infected patients through rapid, simple, but particular and sensitive methods can help prevent the global load on hepatocellular carcinoma (HCC) [4]. Chronic HCV infection progression depends on various factors, which include the age at the time of infection, gender, background, genetic factors of the host, immunity, and viral genotype and subtype [5]. The conversion of HCV into HCC depends on the sex of the HCV-infected individual, comorbidities such as co-infection with HBV or HIV, diabetes, and obesity. The viral genotype (HCV 1b), level of alcohol consumption, and age also play a role as risk factors for the progression of the disease [6]. The leading cause that triggers hepatocellular carcinoma is liver disease. Around seven viral genotypes complicate its treatment [7].

It has been evaluated that around 11 million people in Pakistan are affected with HCV, with almost six circulating subtypes and genotypes of the hepatitis C virus. HCV is viral, and this burden is continuously rising [8]. Hepatitis C virus with chronic and acute infections is responsible for diseases such as hepatocellular carcinoma, cirrhosis, and liver damage. According to the Human Development Index of the United Nations, the ranking of Pakistan is 134th out of 174 countries due to its poor health and educational standards [9].

As such, no vaccine is available yet; the ultimate elimination of this virus requires identifying and treating undiagnosed cases and screening the populations at high risk, such as homosexual men, drug users who inject drugs, and female sex workers [10]. To eliminate HCV, it is critical to improve simple, rapid, and affordable HCV diagnostics [11].

Enzymes such as γ-glutamyl transpeptidase and ALT are specific to the liver. The levels of ALT could also be high in the muscles and kidneys. Alkaline phosphatase and AST enzymes are commonly involved in liver function tests and expressed in bone and muscle [12].

The combination of miRNAs and AFP improves the accuracy of diagnosis compared with AFP or miRNAs alone. The combination of AFP and miRNAs has huge potential as an innovative policy for identifying HCC [13].

A biomarker is also known as a biological remark. It preferably predicts a clinically applicable early outcome and is most difficult to observe. Biomarkers are generally measured in a short period. They could diagnose diseases, screen and monitor predictive indicators, and characterize and develop individualized therapeutic interventions. Biomarkers are also used for treating and predicting adverse drug reactions [14].

MicroRNAs (miRNAs) control the expression of genes post-transcriptionally by attachment to the particular targets of mRNA and promoting their translational inhibition or degradation. MicroRNAs control both pathological and physiological liver functions. A changed miRNAs expression is linked with liver injury, liver metabolism, liver tumor development, and liver fibrosis, making miRNAs attractive as a therapeutic technique for treating and diagnosing liver-related diseases [15]. 

The present study evaluated the different biochemical parameters and viral load in different stages of HCV and the treated group. With the help of bioinformatics tools, key genes were selected, and their expression was seen in different stages of HCV. It has been observed that the gene expression in patients with chronic hepatitis C infection, cirrhosis, and HCC differs from that of normal individuals. Analysis of the gene expression done by real-time PCR is very useful for quantifying the expression of a particular gene in our study. 

## 2. Materials and Methods

### 2.1. Data and Sample Collection

Data were collected after written informed consent was obtained from each patient on the questionnaire. Samples and patients’ clinical histories were collected from different hospitals in Sargodha and Lahore. A questionnaire was designed for the clinical history.

Blood and serum samples were collected from confirmed cases of hepatocellular carcinoma, chronic HCV, and cirrhosis patients who were already confirmed by RT PCR ultrasound of the abdomen. Patients with HIV, HBV, and other viral and bacterial diseases were excluded. Pregnant women; patients on interferon therapy; and patients with autoimmune, inflammatory, and infectious diseases were also excluded—blood samples were used for the isolation of RNA and serum samples were used for determination of the biochemical parameters.

### 2.2. Parameter Analysis

All of the chemical parameters related to the liver were done using a kit method based on spectrophotometry.A commercial ELISA kit determined α fetoproteins.RT PCR determined HCV RNA (viral load).

### 2.3. Expression Analysis

For the expression analysis of selected miRNAs, RNA was extracted from the blood. The RNA quantity was measured with the help of a Nanodrop ND-1000 spectrophotometer. The readings were gained at 260/280 nm.

### 2.4. cDNA Synthesis by Reverse Transcriptase

For the cDNA synthesis, we used a Thermo Scientific RevertAid First cDNA Synthesis Kit. We used a short spin to obtain the homogenous mixture after the addition of all of the components present in the kit. The mixture was placed in a thermo-cycler for incubation at a temperature of 42 °C for 60 min; in this, there were two steps of 30 min each for the reaction of the enzyme, and then incubating it at 70 °C for 5 min to stop the enzyme reaction. We then incubated the mixture at 4 °C for infinity to inactivate the reaction. 

### 2.5. PCR for Expression Analysis of Selected miRNA Genes (miRNA-182, miRNA-122, mi-RNA21, miRNA-199, miRNA-150, and miRNA-222)

The miRNA gene expression was analyzed by cDNA and the specific primers were analysed by BIO-RAD iQ™5 real-time PCR system. The sequence of primers is given in Table 1, according to the instructions of the manufacturer SYBER Green mix. For normalization, the U6 snRNA gene was applied. The control group samples served as calibrators, and their expression levels were fixed to 1. The comparative values of the human mRNAs were normalized against the comparative values of the endogenous control, namely the human U6 snRNA gene. The expression of change in gene fold was calculated by applying the equation 2-ΔΔCT.

Thawed SYBR 2x green Real-time PCR master mix, RNase-free water, and primers were thawed, mixed, and placed over ice. In the end, we added template cDNA to each PCR vessel with the reaction mix. After that, the program of the Real-Time Cycler was selected by following the cycling conditions defined in Figure 1. During the extension step, data acquisition was performed. The comparative expression of the gene analysis was done by using SDS 3.1 software from Bio-Rad Chemical. Every real-time PCR process was carried out in triplicate. The cycling conditions for real-time PCR were 95 °C for 30 s, followed by 40 cycles at 56 °C for 30 s, and 72 °C for 7 min. 

### 2.6. Statistical Analysis

The data were inputted and analyzed using SPSS. The numeric data, such as the biochemical parameters and age, were presented as mean ± S. deviation. Qualitative data such as gender were presented in frequency and percentages. Pearson correlation was used to determine the relationship between different variables. T-test and ANOVA were also used to find differences in the study groups. Chi-square was used to analyze the significant differences among gender and age groups. A *p*-value less than or equal to 0.05 was considered significant. For graphs, graphical representation with the help of Microsoft excel was used.

## 3. Results

### 3.1. Demographics Variables

For this study, patients suffering from hepatitis C with different disease stages, such as chronic (*n* = 40), liver cirrhosis (*n* = 40), and HCC (*n* = 40), were taken. Then, there were two groups of treated (*n* = 40) and control (*n* = 40) individuals. The features of all groups of samples are summarized in Table 2. A sample of 200 individuals was collected, with 40 individuals for each category. A Chi-square test (χ2) was used to find the significant differences in gender and age groups, and a *p*-value at a 0.05 significant level was used to determine the significance. According to the results, no significant difference was found between the male and female ratio, but a significant difference was present between different age groups.

### 3.2. Association Analysis

The association of age, ALT, AST, ALP, viral load, and bilirubin based on gender was tested using a T-test. The complete results were concluded as mean ± standard deviation (SD). The significance of the results was measured by their *p* value (Table 3). A significant difference was present between the age of males and females, but no significant difference was present between the biochemical parameters of the males and females.

### 3.3. Analysis of Variance

The variance of age and biochemical parameters such as ALT, AST, ALP, viral load, and bilirubin in patients with hepatitis C at different disease stages, such as chronic, cirrhosis, and HCC, as well as for the treated and control, was calculated using ANOVA. The significance of the results was measured by their *p* value (Table 4 and Figure 2). A significant difference was present between the biochemical parameters and age of the diseased groups and control, except for the viral load, which was not significant.

### 3.4. Association Analysis

The association of age, ALT, AST, ALP, viral load, and bilirubin considering the chronic and treated groups was tested using a T-test. The results were concluded as mean ± standard deviation (SD). The significance of the results was measured by their *p* value (Table 5) and Figure 3. A significant difference was observed between the age of the chronic and treated groups, but other biochemical parameters showed a non-significant difference.

### 3.5. Expression Analysis of miRNAs in Diseased and Control Groups Using Real-Time PCR

The expression of miRNA 182, miRNA -122, miRNA -21, miRNA -199, miRNA -150, and miRNA-222 was measured by RT-PCR with the help of specific primers for the gene and SYBR Green mix. For normalization, U6 snRNA was used. Each PCR reaction was performed in triplicate in real time.

For the expression of miRNA 182, an almost 2.3, 2.5, and 3.8 fold increased expression was observed in the chronic, cirrhosis, and HCC samples, respectively. For the expression of miRNA 122, almost 2.8, 2.7, and 4.5 fold increased expression was observed in chronic and HCC samples, respectively, but the expression of miRNA 122 decreased in the cirrhosis compared with chronic group. In the chronic group, the expression was 2.8 fold, while in cirrhosis, it was 2.7. For the expression of miRNA-21, an almost 3.5, 3.8, and 5.22 fold increased expression was observed in the chronic, cirrhosis, and HCC samples, respectively. For the expression of miRNA-199, an almost 3.2 and 4.7 fold increased expression was observed in the chronic and HCC samples, respectively, but the expression of miRNA-199 decreased in the cirrhosis compared with chronic group. In the chronic group, the expression was 3.2 fold, while in cirrhosis, it was 3 fold. The expression of miRNA-150 was almost 3 fold; increased expression was observed in the chronic group but the expression of miRNA-150 was decreased in the cirrhosis and HCC groups. In the cirrhosis group, the expression was 2 fold; in HCC, it was 1.5 fold compared with the chronic. The expression of miRNA-222 was increased by almost 3.8 and 5.6 fold in the chronic and HCC samples, respectively, but the expression of miRNA-222 decreased in the cirrhosis compared with chronic group. In the chronic group, the expression was 3.8 fold, while in cirrhosis, it was 3. The results are presented graphically in order to make a clear comparison among the different fold expressions of the disease groups compared with the control group, Figure 4.

### 3.6. Expression Analysis of miRNA in Chronic Patient and Treated Groups Using Real-Time PCR

The expression of miRNA 182, miRNA -122, miRNA -21, miRNA -199, miRNA -150, and miRNA-222 was quantified by RT-PCR with the help of specific primers and SYBR Green mix. For normalization, U6 snRNA was used. Each PCR reaction in real time was performed in triplicate.

Compared with the diseased group (HCV), the miRNA expression was decreased in the treated group as it was 0.5 fold in miRNA-182, 0.4 fold in miRNA-122, 0.35 fold in miRNA-21, 0.36 fold in miRNA-199, 0.45 fold miRNA-150, and 0.6 fold miRNA-222. The results are graphically represented in Figure 5.

## 4. Discussion

Hepatitis C virus is commonly asymptomatic, so detection is very difficult in its early stage. HCV is considered a “silent killer” because patients do not receive treatment in the early stage [16].

In the present study, six miRNAs, miRNA-182, 199, 221, 122, 21, and 150, were selected for quantification of the expression using real-time PCR. This study comprised five groups. Group 1 was chronic, group 2 was cirrhosis, group 3 was HCC, group 4 comprised of healthy controls, and group 5 comprised of treated patients. We conducted comparisons between the three diseased groups and normal controls, and the HCV group and treated patients.

Enzymes such as γ-glutamyl transpeptidase (GGT) and ALT are considered liver-specific. The levels of ALT could be high in the muscles or kidneys. Alkaline phosphatase and AST enzymes are commonly used in liver function tests and expressed in bone and muscle, respectively [12]. Alanine transaminase (ALT), aspartate transaminase (AST), and α-fetoprotein (AFP) levels in the serum are considered as biomarkers for the diagnosis of HCV. When diagnosing HCC, the α-fetoproteins (AFP) levels play an important role.

Serum levels of the liver enzymes ALT and AST are high during liver inflammation. In the current study, the serum levels of ALT and AST were higher in the HCC patients than in the chronic and normal individuals. The mean ± SD of ALT in HCC was 43.08 ± 20.42, while in the chronic it was 35.38 ± 15.70; in cirrhosis, it was 35.13 ± 12.38, and in the controls, it was 29.05 ± 3.81. Similarly, the mean ± SD of AST in HCC was 41.65 ± 21.83 compared with chronic, which was 35.38 ± 15.70; in the cirrhosis group, it was 32.90 ± 12.74, and in the normal controls, it was 27.88 ± 3.04. In comparison, there were statistically significant results with the *p* values (*p* ≤ 0.001). Neamatallah et al. (2014) found similar results compared with the present study with ALT 51.2 ± 2.4 and AST 57.6 ± 3.6 cases of HCC [17]. Sheth et al. (1998) described the opposite results in which ALT and AST levels were used to distinguish chronic and cirrhosis patients. The mean values of ALP, AST, and ALP were significant between the studied groups, *p* = 0.000 [18].

The mean value of bilirubin was high in HCC compared with the other groups, and was 1.65 ± 0.75, so this parameter could be indicative of HCC. Data regarding the high mean value of bilirubin in the HCC compared with HCV and cirrhosis were in agreement with Carr et al. (2014) [19]. They reported a high level of bilirubin in HCC. The results of the study of Saini et al. (2006) were also according to our findings related to high levels of ALT, AST, ALP, and bilirubin in HCC patients. In a recent study, the viral load was higher in chronic patients compared with those with cirrhosis and HCC [20].

The present study’s age-wise comparison revealed that the eldest patients belonged to the HCC group, with a mean ± SD 56.85 ± 6.62. In the present study, HCC was more prevalent in males compared with females, where out of 40 patients, 10 were female and 30 were male. Acharya’s (2014) results support these findings related to the age and sex of HCC patients [21]. They reported that age was between 40 to 70 years and the male-to-female ratio for HCC in India was 4:1. According to the study of Lou et al. (2013), the mean age of HCC patients was 53.6, in agreement with the present findings [22]. This shows that an age above 50 could be a risk factor for tumor development in HCV patients.

The present study revealed that HCC patients had a high range of AFP with 808.15 ± 1381.09. The study of Saini et al. (2006) supports these findings, with up to 400 ng/mL AFP level in HCC [20]. MicroRNAs (miRNAs) target the critical host factors needed for enlarged cell growth and productive HCV replication, and play an essential role in the establishment of HCV infection. The altered expression of microRNAs is involved in the pathogenesis associated with HCV infection by using control signaling pathways such as proliferation, immune response, and apoptosis [23].

Li et al. (2019) found four miRNAs (miR-122-5p, miR-331-3p, miR-494-3p, and miR-224-5p) were considerably increased, and two miRNAs (miR-185-5p and miR-23b-3p) were considerably decreased in HCC patients compared with LC patients [24]. These results contradict our recent findings, where the expression of micro RNA 122 decreased in LC compared with HCC. Shaheen et al. (2018) reported a significant decrease in miR-182 and miR-150 in HCC compared with HCV patients, disagreeing with the present study where miRNA-150 expression was decreased in HCC but miRNA-182 expression was increased [25]. Khairy et al. (2021) measured the serum levels of four miRNAs (miRNA-21, 199, 448, and 181c) by using quantitative real-time PCR assay; the results showed upregulation of these miRNAs; these results are in support of the present study as the level of miRNA 21 and 99 increased in HCV [26]. Nasser et al. (2019) also favored the present study in miRNA-21 [27]. Bihrer et al. (2011) also reported that MicroRNA-21 (miR-21) was upregulated in the tissue of tumors in patients with malignant diseases, which also included hepatocellular carcinoma (HCC) [28].

El-Abd et al. (2015) reported that the serum level of miR-199a in chronic HCV patients was considerably lower than in healthy controls, opposing the present study’s results in which the miRNA-199 expression increased in HCV compared with the control and as the disease progressed [29]. El-Guendy et al. (2016) demonstrated that miR-21, miR-122, and miR-199a were down-regulated in the case of HCV compared with the control [30]. These results contradict our findings in that these miRNAs were upregulated in HCV compared with the control. Dubin et al. (2014) reported that Chronic-HCV patients had very elevated serum miR-122 levels in the range of most patients with severe hepatic injury leading to acute liver failure [31]. These results are in favor of the recent study. miR-221 has been shown to play a crucial role in the development of liver fibrosis, a common feature of most types of liver diseases [32]. Bala et al. (2012) found that the serum levels of miR-122 were elevated in HCV patients and were correlated with increased ALT and AST levels and serum miR-155 levels; also, in support of the present study, as in the present results, there was the elevation of miRNA 221 in HCV compared with the controls [33].

## 5. Conclusions

The present study was conducted to determine the differently expressed genes at the miRNA level and to quantify their expression using real-time PCR. Our study comprised five studied groups, namely the chronic HCV group, cirrhosis group, HCC group, control group, and treated groups. Biochemical parameters such as ALT, AST, ALP, and bilirubin were also determined. Along with the biochemical parameters, viral load, and AFP in the case of HCC were also determined. The value of AFP was high in HCC patients, so it could be used as a diagnostic biomarker for HCC; it was not raised in all patients, so we cannot rely only on AFP. The association of age, ALT, AST, ALP, viral load, and bilirubin based on gender was tested using a T-test. The variance of age, ALT, AST, ALP, viral load, and bilirubin in patients with hepatitis C of different disease stages such as chronic, cirrhosis, and HCC, as well as treated patients and the control group was calculated using ANOVA. Their *p* value measured the significance of the results. A comparison was made between the HCV and treated groups. The viral load was high in chronic patients, but the viral load was below the detectable level after successful treatment. MiRNAs are differentially expressed in HCV and as the disease progresses, so these can be used as potential biomarkers for the diagnosis of HCC and other stages and treatment strategies. Six genes at the miRNA level, i.e., miRNA 182, miRNA -122, miRNA -21, miRNA -199, miRNA -150, and miRNA-222, were selected, and the expressions of these genes were quantified by real-time PCR. These genes were differentially expressed in all of the studied groups when we compared them with the control. Expression of all miRNAs was elevated in all of the studied groups. All of these miRNAs showed a decreased expression in the treated group because of successful treatment.

## Figures and Tables

**Figure 1 genes-14-00441-f001:**
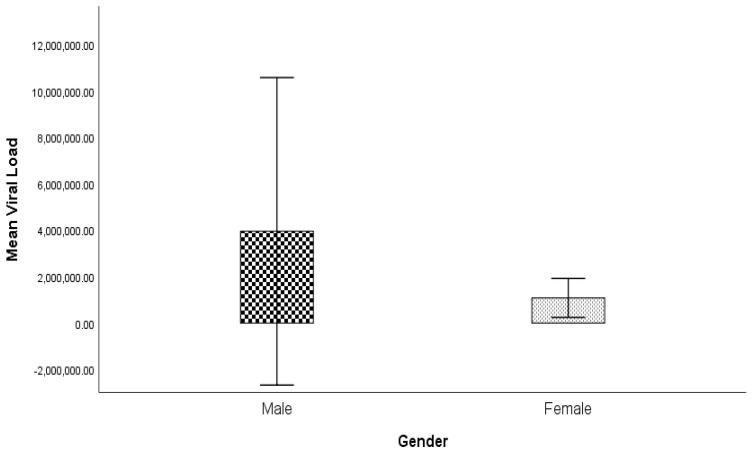
Graphical representation of comparison of viral load among males and females.

**Figure 2 genes-14-00441-f002:**
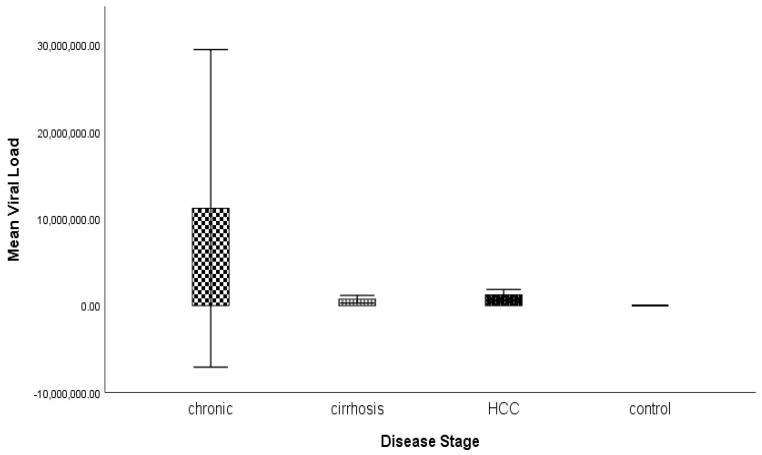
Graphical representation of analysis of variance in viral load level among the chronic, cirrhosis, HCC, treated, and control groups.

**Figure 3 genes-14-00441-f003:**
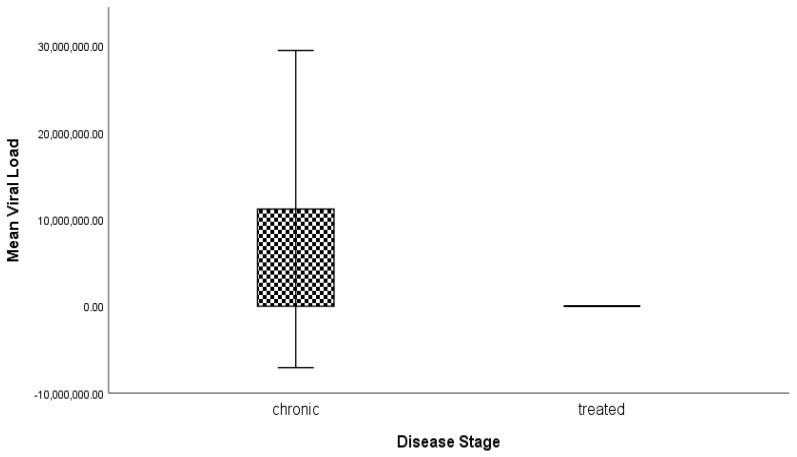
Graphical representation of the comparison of viral load among the chronic and treated groups.

**Figure 4 genes-14-00441-f004:**
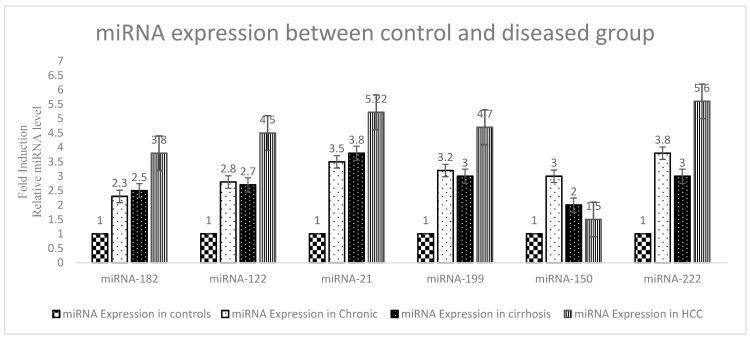
Graphical representation of expression analysis of miRNAs in diseased and control groups using real-time PCR.

**Figure 5 genes-14-00441-f005:**
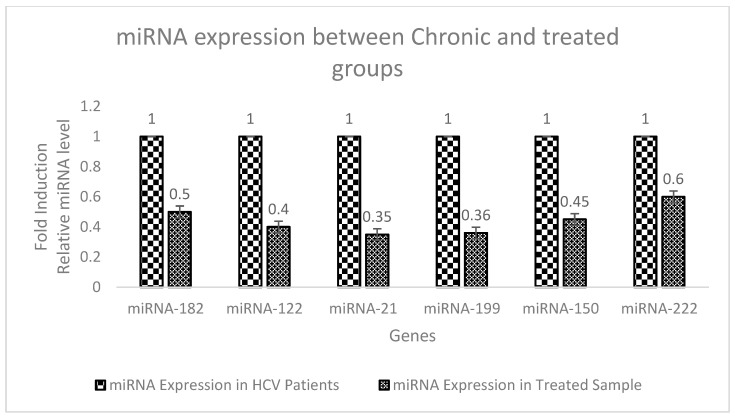
Graphical representation of the expression analysis of miRNAs in the diseased and treated groups using real-time PCR.

**Table 1 genes-14-00441-t001:** Primers of miRNAs (miRNA-182, miRNA-122, miRNA-21, miRNA-199, miRNA-150, and miRNA-222).

Primer Name	Primer Sequence
*miRNA-182-F*	5′-TGCGGTTTGGCAATGGTAGAAC-3’
*miRNA-182-R*	5’-CCAGTGCAGGGTCCGAGGT-3’
*miRNA-122–F*	5′-ACACTCCAGCTGGGTGGAGTGTGACAATCC-3′
*miRNA-122R*	5′-TGGTGTCGTGGAGTCG-3′
*miRNA-21-F*	5′-GCCCGCTAGCTTATCAGACTGATG-3′
*miRNA-21-R*	5′-CAGTGCAGGGTCC GAGGT-3
*miRNA-199–F*	5′-GCGGCGGACAGTAGTCTGCAC-3′
*miRNA-199–R*	5′-ATCCAGTGCAGGGTCCGAGG-3′
*miRNA-150-F*	5′TCTCCCAACCCTTGTACCAGTG3′
*miRNA-150-R*	5′CAGTGCGTCGTGGAGT3′
*miRNA-222-F*	5′-GTTCGTGGGAGCTACATTGTCTGC-3′
*miRNA-222-R*	5′-GTGTCGTGGAGTCGGCAATTC-3′
*U6 snRNA-F*	CTCGCTTCGGCAGCACATATAC
*U6 snRNA-R*	ACGCTTCACGAATTTGCGTGTC

**Table 2 genes-14-00441-t002:** Frequency distribution of all demographics for the patients and control subjects.

Variables		Chronic (*n* = 40)	Cirrhosis (*n* = 40)	HCC (*n* = 40)	Treated (*n* = 40)	Control (*n* = 40)	*p* Value
**Gender**	**Male**	20 (50%)	20 (50%)	30 (75%)	18 (45%)	20 (50%)	0.057
	**Female**	20 (50%)	20 (50%)	10 (25%)	22 (55%)	20 (50%)	
**Age**	**21–40**	22 (55%)	8 (20%)	1 (2.5%)	10 (25%)	30 (75%)	**0.001**
	**41–60**	15 (37.5%)	29 (72.5%)	29 (72.5%)	26 (65%)	6 (15%)	
	**61–80**	3 (7.5%)	3 (7.5%)	10 (25%)	4 (10%)	4 (10%)	

Note: Bold values are significant.

**Table 3 genes-14-00441-t003:** Association of age and biochemical parameters based on gender.

	Male (*n* = 108)	Female (*n* = 92)	*p* Value
**Variables**	**M ± SD**	**M ± SD**	
**Age**	48.96 ± 11.96	44.48 ± 10.65	0.006
**ALT**	35.18 ± 17.50	32.48 ± 11.08	0.20
**AST**	34.43 ± 15.69	31.34 ± 10.74	0.11
**ALP**	175.05 ± 105.46	157.52 ± 57.62	0.16
**Viral Load**	3,952,923.66 ± 34,763,237.74	1,081,468.79 ± 4,053,847.94	0.43
**Bilirubin**	1.01 ± 0.60	0.88 ± 0.55	0.11

Note: Bold values are significant.

**Table 4 genes-14-00441-t004:** Analysis of variance of age and biochemical parameters in the chronic, cirrhosis, HCC, treated, and control groups.

Variables	Chronic (*n* = 40)	Cirrhosis (*n* = 40)	HCC (*n* = 40)	Control (*n* = 40)	*p* Values
Age	41.35 ± 11.55	49.08 ± 9.11	56.85 ± 6.62	38.50 ± 10.86	0.000
ALT	35.38 ± 15.70	35.13 ± 12.38	43.08 ± 20.42	29.05 ± 3.81	0.000
AST	33.13 ± 12.50	32.90 ± 12.74	41.65 ± 21.83	27.88 ± 3.04	0.000
ALP	179.53 ± 45.38	163.70 ± 68.85	210.85 ± 137.64	91.65 ± 19.83	0.000
Viral Load	11,191,265.55 ± 57,129,256.41	735,073.68 ± 1,357,141.14	1,233,932.88 ± 1,946,918	0.00 ± 0.00	0.25
Bilirubin	0.77 ± 0.26	0.94 ± 0.48	1.65 ± 0.75	0.64 ± 0.20	0.000
AFP			808.15 ± 1381.09		NA

**Table 5 genes-14-00441-t005:** Association of age and biochemical parameters considering the chronic and treated groups.

	Chronic (*n* = 40)	Treated (*n* = 40)	*p* Value
**Variables**	**M ± SD**	**M ± SD**	
**Age**	41.35 ± 11.55	48.73 ± 9.52	**0.003 ****
**ALT**	35.38 ± 15.70	31.63 ± 4.67	0.15
**AST**	33.13 ± 12.50	29.50 ± 5.98	0.10
**ALP**	184.48 ± 35.17	189.20 ± 63.56	0.68
**Viral Load**	11,191,265.55 ± 57,129,256.41	0.00 ± 0.00	0.22
**Bilirubin**	0.77 ± 0.26	0.77 ± 0.39	0.95

Note: Bold values are significant. ** Significant at the 0.01 level.

## Data Availability

Not applicable.

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
