# Peer review of "Serum MicroRNAs as Predictors for HCV Progression and Response to Treatment in Pakistani Patients"

_genes, 2023, doi:10.3390/genes14020441_

Round 1

Reviewer 1 Report

The topic of the manuscript is interesting and relevant. The study design is well done. The author,s writing style is scientific and analytical. The obtained results regarding miRNA will have application in practice. I have objections to the fact that the same expressions are present in both - the introduction and  the discussion, which is unnecessary. It is necessary to remove from the discussion  the general information, which is also present in the introduction.

Author Response

Author's Reply to the Reviewer #1

Major Comments

  1. I have objections to the fact that the same expressions are present in both - the introduction and the discussion, which is unnecessary. It is necessary to remove from the discussion the general information, which is also present in the introduction.

Response) We are grateful to the reviewer for a detailed review and agree with the reviewer’s point, so the unnecessary information is deleted from the discussion. The deleted lines are given below

The important pathogen in humans that is responsible for causing HCC, liver cirrhosis, and hepatitis is the Hepatitis C virus (HCV). Dubuisson and Cosset, (2014). According to World Health Organization records, about 71 million people are infected 235 with the Hepatitis C virus worldwide. It is reported that around 399,000 people are 236 infected with the Hepatitis C virus converts to failure of the liver, cirrhosis, and hepatocellular carcinoma every year Warkad et al., (2018).

Reviewer 2 Report

Line 57-64:The sentences and references are similar. As a result, please rewrite them in your own sentences.

Figure 1 and Figure 2 : Both of them could be written in the manuscript as sentences.

Table 1: The primers for miRNA 222-F that you provide in the table are similar to miRNA-221 in the miRNA database.
If you designed your research for 221 please rewrite the manuscript or correct/rewrite again according to your goal.

Table 2,3,4,5: Tables are difficult to understand, so please redesign the tables and explanations.

Figure 4: The various figures could be represented by a line chart, which is preferable to a bar chart. Because 1 number is stable in your figure.

Author Response

Author's Reply to the Reviewer #2

Major Comments

  1. Line 57-64: The sentences and references are similar. As a result, please rewrite them in your own sentences.

Response) According to Reviewer’s suggestion, sentences have been rephrased as

It is evaluated that around 11 million people in Pakistan are affected with HCV, with almost six circulating subtypes and genotypes of the hepatitis C virus. HCV is viral, and this burden is continuously rising. (Zafar et al., 2018). Hepatitis C virus with chronic and acute infections is responsible for diseases such as hepatocellular carcinoma, cirrhosis, and liver damage. According to the Human Development Index of the United Nations, the ranking of Pakistan is 134th of 174 countries due to its poor health and educational standards (Arshad & Ishfaq, 2017).

  1. Figure 1 and Figure 2: Both of them could be written in the manuscript as sentences.

Response) We are grateful to the reviewer for a detailed review and agree with the reviewer’s suggestion, the figures 1 and figure 2 have been removed and describe the process of cDNA synthesis and Real-time PCR protocol in sentences as given below,

Explanation of figure 1: it was a short spin to get the homogenous mixture after the addition of all components present in the kit. The mixture was placed in a thermo-cycler for incubation at a temperature of 42 °C for 60 minutes, in this, there were two steps of 30 min each for the reaction of the enzyme, to incubate it at 70 °C for 5 min to stop the enzyme reaction. Incubated the mixture at 4 °C for infinity to inactivate the reaction.

Explanation of figure 2: 95Ëš C for 30 seconds, followed by 40 cycles at 56Ëš C for 30 seconds, and 72Ëš C for 7 minutes.

  1. Table 1: The primers for miRNA 222-F that you provide in the table are similar to miRNA-221 in the miRNA database.
If you designed your research for 221 please rewrite the manuscript or correct/rewrite again according to your goal.

Response):  The primers used for miRNA 222 supported by the research paper as given below,

The reference paper for miRNA 222-F primer is Xiang et al., 2019 (miR-222 expression is correlated with the ATA risk stratification in papillary thyroid carcinomas).

  1. Table 2,3,4,5: Tables are difficult to understand, so please redesign the tables and explanations.

Response) Extra information is deleted from tables and explanations of tables are also revised according to reviewers’ instructions.

  1. Figure 4: The various figures could be represented by a line chart, which is preferable to a bar chart. Because 1 number is stable in your figure.

Response) We are grateful to the reviewer for a detailed review, but bar charts are better than line charts to compare the expression of miRNAs at different stages of HCV. All literature that we got from different research papers also compared different miRNAs with the help of a bar chart.

Reviewer 3 Report

1.       “It is reported that around 399,000 people identified with the Hepatitis C virus convert to liver failure, cirrhosis, and hepatocellular carcinoma (HCC) yearly.” What is the disease distribution look like depending on gender, age, economic background etc.?

2.       Please elaborate on the roles of miRNA in physiological and/or pathological conditions.

3.       A 3-4-fold change in mRNA expression is not very significant. Significance/non-significance marks are missing on the graphs.

4.       Providing viral load data/graphs would be nice and add to better visual comprehension.

5.       “Li et al. (2019) found four miRNAs (miR-122-5p, miR-331-3p, miR-494-3p, miR-224-5p) 284 were considerably increased, and two miRNAs (miR-185-5p, miR-23b-3p) were consid- 285 erably decreased in HCC patients as compared to LC patients. These results contradict 286 our recent findings where expression of micro RNA 122 decreased in LC compared to 287 HCC. Shaheen et al. (2018) reported a significant decrease of miR-182 and miR-150 in 288 HCC compared to HCV patients disagreeing with the present study where miRNA-150 289 expression is decreased in HCC but miRNA-182 expression increased.” What could be the possible reasons for the contradiction in results?

6.       “ El-Abd et al. (2015) reported that the serum level of miR-199a in chronic HCV patients 298 was considerably lower than in healthy controls, opposing the present study's results in 299 which miRNA-199 expression increased in HCV as compared to control and as the dis- 300 ease progressed. El-Guendy et al. (2016) demonstrated that miR-21, miR-122, and miR- 301 199a were down-regulated in the case of HCV compared to the control. These results 302 contradict our findings in that these miRNAs are upregulated in HCV compared to the 303 control.” Please elaborate on molecular level the possible reasons for the contradictions.

Author Response

Author's Reply to the Reviewer #2

Major Comments

  1. Line 57-64: The sentences and references are similar. As a result, please rewrite them in your own sentences.

Response) According to Reviewer’s suggestion, sentences have been rephrased as

It is evaluated that around 11 million people in Pakistan are affected with HCV, with almost six circulating subtypes and genotypes of the hepatitis C virus. HCV is viral, and this burden is continuously rising. (Zafar et al., 2018). Hepatitis C virus with chronic and acute infections is responsible for diseases such as hepatocellular carcinoma, cirrhosis, and liver damage. According to the Human Development Index of the United Nations, the ranking of Pakistan is 134th of 174 countries due to its poor health and educational standards (Arshad & Ishfaq, 2017).

  1. Figure 1 and Figure 2: Both of them could be written in the manuscript as sentences.

Response) We are grateful to the reviewer for a detailed review and agree with the reviewer’s suggestion, the figures 1 and figure 2 have been removed and describe the process of cDNA synthesis and Real-time PCR protocol in sentences as given below,

Explanation of figure 1: it was a short spin to get the homogenous mixture after the addition of all components present in the kit. The mixture was placed in a thermo-cycler for incubation at a temperature of 42 °C for 60 minutes, in this, there were two steps of 30 min each for the reaction of the enzyme, to incubate it at 70 °C for 5 min to stop the enzyme reaction. Incubated the mixture at 4 °C for infinity to inactivate the reaction.

Explanation of figure 2: 95Ëš C for 30 seconds, followed by 40 cycles at 56Ëš C for 30 seconds, and 72Ëš C for 7 minutes.

  1. Table 1: The primers for miRNA 222-F that you provide in the table are similar to miRNA-221 in the miRNA database.
If you designed your research for 221 please rewrite the manuscript or correct/rewrite again according to your goal.

Response:  The primers used for miRNA 222 supported by the research paper as given below,

The reference paper for miRNA 222-F primer is Xiang et al, 2019 (miR-222 expression is correlated with the ATA risk stratification in papillary thyroid carcinomas).

  1. Table 2,3,4,5: Tables are difficult to understand, so please redesign the tables and explanations.

Response) Extra information is deleted from tables and explanations of tables are also revised according to reviewers’ instructions.

  1. Figure 4: The various figures could be represented by a line chart, which is preferable to a bar chart. Because 1 number is stable in your figure.

Response) We are grateful to the reviewer for a detailed review but bar charts are better than line charts to compare the expression of miRNAs at different stages of HCV. All literature that we got from different research papers also compared different miRNAs with the help of a bar chart.

Round 2
